# Minimizing Energy Consumption Leads to the Emergence of Gaits in Legged Robots

**Zipeng Fu[1],  Ashish Kumar[2],  Jitendra Malik[2],  Deepak Pathak[1]**
[1]Carnegie Mellon University,   [2]UC Berkeley

**Abstract:** Legged locomotion is commonly studied and expressed as a discrete set of gait patterns, like walk, trot, gallop, which are usually treated as given and pre-programmed in legged robots for efficient locomotion at different speeds. However, fixing a set of pre-programmed gaits limits the generality of locomotion. Recent animal motor studies show that these conventional gaits are only prevalent in ideal flat terrain conditions while real-world locomotion is unstructured and more like bouts of intermittent steps. What principles could lead to both structured and unstructured patterns across mammals and how to synthesize them in robots? In this work, we take an analysis-by-synthesis approach and learn to move by minimizing mechanical energy. We demonstrate that learning to minimize energy consumption plays a key role in the emergence of natural locomotion gaits at different speeds in real quadruped robots. The emergent gaits are structured in ideal terrains and look similar to that of horses and sheep. The same approach leads to unstructured gaits in rough terrains which is consistent with the findings in animal motor control. We validate our hypothesis in both simulation and real hardware across natural terrains. Videos at https://energy-locomotion.github.io.

**Keywords:** Locomotion, Biomechanics, Energetics, Reinforcement Learning

## 1   Introduction

A common approach to understanding mammalian locomotion is to study it in terms of a set of discrete gaits. Seminal works include Muybridge [1]'s motion video of a galloping horse and Hildebrand [2]'s study of different gaits observed at different speeds. This has inspired research on replicating these gaits in robots by pre-programming them to achieve effective locomotion. This approach of pre-programming fixed gait patterns has lead to immense progress in robotic locomotion [3, 4, 5, 6].

However, pre-programmed gaits limit the ability of legged systems to perform general locomotion in diverse terrains and at different speeds. Studies of motor control in both animals and human infants [7] in the last few decades have shown that general locomotion in natural settings is irregular and does not fit predefined fixed gaits. It is better defined in terms of bouts of steps than periodic cycles [8, 9]. That being said, we do see consistent regular patterns in ideal conditions when an animal is moving in a straight line over flat terrain at different speeds, e.g., walk, trot, canter, gallop, etc. [2, 10, 11, 12] and these patterns are shared across species of diverse animals [13]. How do we reconcile these seemingly contradictory arguments about unstructured locomotion patterns on complex terrains with that of regular patterns seen in flat terrains under the same rubric?

One way to make sense of these differences is to see them from the lens of minimizing energy consumption. Locomotion consumes a significant fraction of an animal's metabolic energy budget which suggests that there would be evolutionary selection pressure for locomotion strategies that minimize energy consumption. Indeed, this has been a focus of several studies in biomechanics and energetics. Hoyt and Taylor [12] used energy minimization to explain why animals switch from one gait to another as their speed increases. Similar studies explain gaits across different animals including humans [14, 15]. However, most of the biomechanics studies are limited to the straight-line motion of animals in ideal flat terrains. Furthermore, these studies take gaits as given and offer simplified analytical models to explain them from an energy perspective without any statement about their synthesis on relatively complex real robot systems.

5th Conference on Robot Learning (CoRL 2021), London, UK.

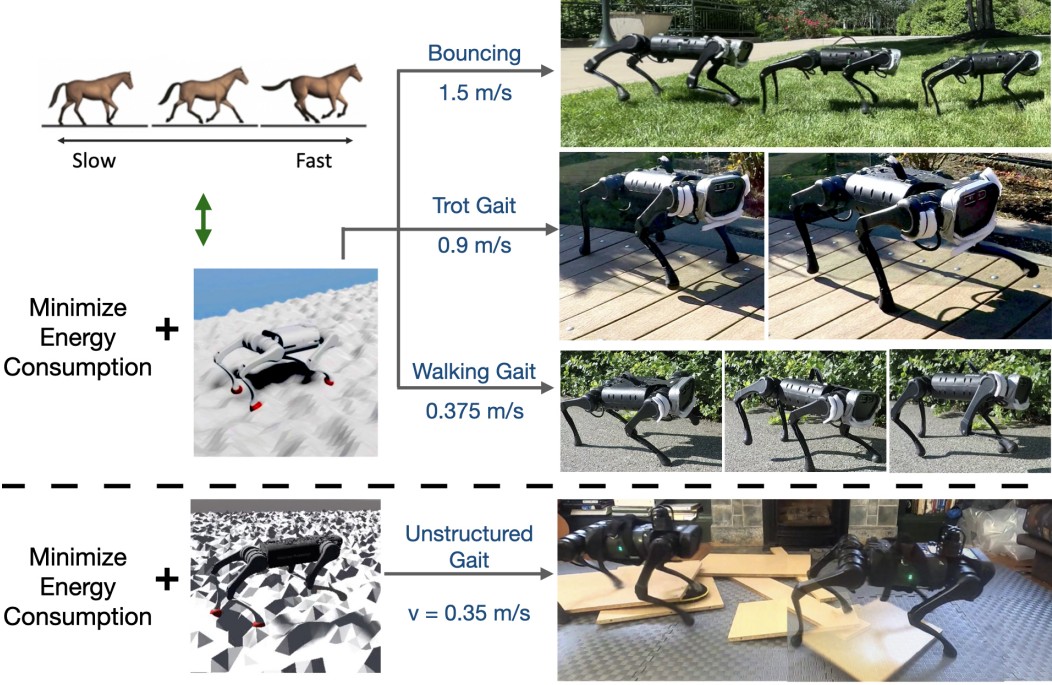

Figure 1: We demonstrate via analysis-by-synthesis approach that learning to move forward by minimizing energy consumption plays a key role in the emergence of natural locomotion patterns in quadruped robots. We do not pre-program any primitives for leg motions and the policy directly output desired joint angles. `Top`: Bio-energetics driven learning on flat terrain leads to different gaits namely walk, trot and bounce (similar to gallop) as the speed increases. Our high-speed bounce gait displays an emergent *flight* phase despite low energy usage. This corresponds to the nearest animals (sheep/horse) with similar Froude numbers. `Bottom`: The same pipeline on diverse uneven terrains leads to unstructured gait as is true in general animal locomotion. Please see videos on the website.

In this work, we take an analysis-by-synthesis approach to show how energy minimization leads to the emergence of structured locomotion gait patterns in flat terrains as well as unstructured gaits in complex terrains. We use energetics to design an end-to-end learning framework and display the resulting gait patterns in a real quadruped robot. We employ model-free reinforcement learning as the optimizer to learn controllers to make the quadruped move forward while simultaneously minimizing the bio-energetics constraints of minimizing work [16]. We train our policies on simple fractal terrains with varying frequency of terrain heights instead of perfectly flat terrain. This mimics the real world more closely and achieves efficient and robust gaits without the need for artificial rewards for foot clearance or periodic external push during training. Low frequency fractal terrains resemble the simple flat terrain settings and high frequency simulates complex terrains. In addition, we present a learning pipeline that enables smooth gait transitions given changing target speeds by combining RL rewards and distillation losses from trained expert gait policies. We then transfer the policies onto the real robot using online adaptation at test time [17].

We further analyze our emergent walking behaviors by comparing them to biological quadrupeds. Our quadruped robot is closest to sheep and horse in terms of physicality as determined by the Froude number [18] which is a scalar metric characterizing gaits of quadrupeds (discussed more in Section 4.2). Therefore, energetics-based training leads to similar three gaits in our robot as found in horses and sheep, namely, walk, trot and gallop as the speed is gradually increased in flat terrains Figure 1. Interestingly, we show that the gait selection is inherently linked with the speed of the robot. For instance, trotting is only energy efficient around the medium speed of 0.9m/s. If we increase or decrease the speed of the robot, it takes more energy for trot than to gallop (similar to bouncing gait) or walk as shown in Figure 2. Finally, when the same pipeline is run in uneven complex terrains, it leads to unstructured gaits as consistent with animal locomotion in usual diverse conditions [9, 8].

Prior works have used energy penalty to obtain energy-efficient gaits [19, 20] but have not shown the correspondence to different gaits at different speeds. Furthermore, out of these works, the ones which deploy these gaits on a real robot either use a predefined hand-coded gait library, use reference motions [19], or only learn high-level contact sequences for a predefined swing leg motion [21].

These approaches have some fundamental limitations in contrast to what we propose. Firstly, the use of predefined foot motions never optimize for efficient swing leg motions, and using reference trajectories gives gaits that look realistic, but are not necessarily energy-efficient for the specific robot we have. Second, learning controllers end to end without the use of any priors (reference motions or predefined motions) allows for the possibility of learning behaviors that don't follow any predefined gaits. We show the robustness of this complex terrain policy by successfully deploying it in complex terrains in the real world.

The main contributions of this paper include:

- Show that minimizing energy consumption plays a key role in the emergence of natural locomotion patterns in both flat as well as complex terrains at different speeds without relying on demonstrations or predefined motion heuristics.
- Show that the emergent gaits at different target speeds correspond to conventional animals in the similar Froude number range (sheep/horse) without any sort of pre-programming.
- Present a distillation-based learning pipeline to obtain velocity-conditioned policy that displays smooth gait transition as the target speed is changed.
- Demonstrate the emergent behaviors, robustness analysis, and gait patterns in simulation as well as a real-world budget quadruped robot.

## 2 Method

### 2.1 Claim: Energy Minimization leads to Emergence of Natural Gaits

We would like to validate that indeed certain gaits will emerge at certain speeds if we minimize the energy consumption. To do so, on the right, we plot the energy consumption vs speed of the A1 robot in simulation for walk, trot and bounce (modification of gallop/canter) gaits. We use [6] to achieve the different gaits by hard-coding the contact sequences for walk, trot and bounce. This plot is qualitatively similar to the plot in [12], which shows the energy consumption of different horse gaits across different speeds. We additionally plot the energy consumption of the three policies we get from our approach at different speeds (0.375 m/s, 0.9 m/s, 1.5 m/s). This plot shows two things. First, it is observed that certain gaits are only efficient at certain speeds and using them for other speeds is suboptimal. For instance, walk is optimal at low speeds with transition to trot and bounce. Secondly, we observe that our learning framework converges to the optimal gait at each of the speeds, and beats the efficiency of MPC gaits

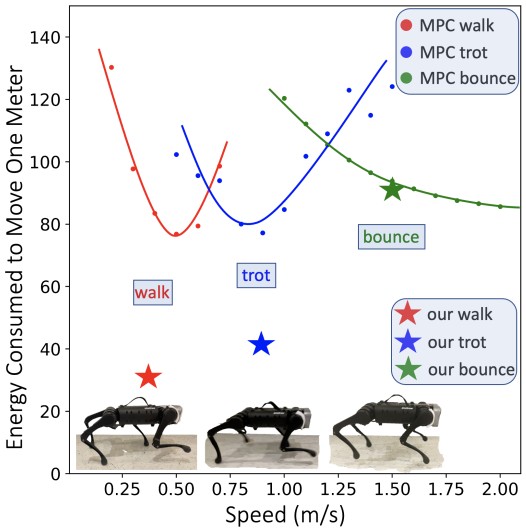

Figure 2: Energy consumed in moving 1m distance using our method and MPC shows why certain gaits are stable at certain speeds. Videos on the website.

from [6], which does not optimize the swing foot trajectory for energy. Details of MPC method are in the supplementary.

### 2.2 Learning Locomotion Policy

We learn the locomotion policy $\pi$ which takes as input the current state $x_t \in \mathbb{R}^{30}$, previous action $a_{t-1} \in \mathbb{R}^{12}$ to predict the next action $a_t$ (Equation 1). The predicted action $a_t$ is the desired joint position for the 12 robot joints which is converted to torque using a PD controller.

$$a_t = \pi(x_t, a_{t-1}) \tag{1}$$

We implement $\pi$ as MLPs (details in Section 3) and train the base policy $\pi$ end to end using model-free reinforcement learning. At time step $t$, $\pi$ takes the current state $x_t$, previous action $a_{t-1}$ to predict an action $a_t$. RL maximizes the following expected return of the policy $\pi$:

$$J(\pi) = \mathbb{E}_{\tau \sim p(\tau|\pi)} \left[ \sum_{t=0}^{T-1} \gamma^t r_t \right], \tag{2}$$

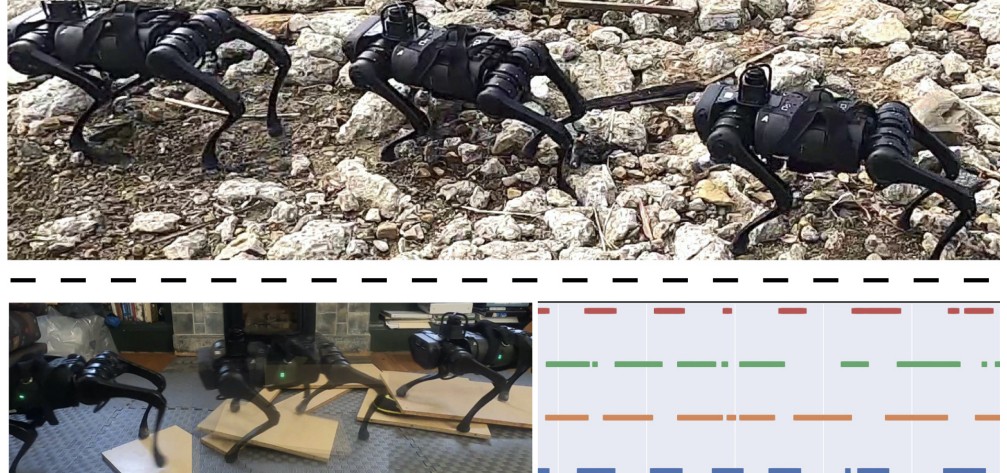

Figure 3: Complex real-world behaviors at the low speed in 2 settings. `Top:` key frames on rocky terrain. `Bottom:` Foot contact plots and key frames on unstable moving planks. Videos on the website.

where $\tau = \{(x_0, a_0, r_0), (x_1, a_1, r_1)...\}$ is the trajectory of the agent when executing policy $\pi$, and $p(\tau|\pi)$ represents the likelihood of the trajectory under $\pi$.

## 2.3  Stable Gait through Natural Constraints

We train our agent with the following natural constraints. First, the reward function is motivated from bio-energetic constraints of minimizing work [16]. We found these reward functions to be critical for learning realistic gaits in simulation. Second, we train our policies on uneven terrain (Figure 1) as a substitute for additional rewards used by [22] for foot clearance and robustness to external push. We see that these constraints are enough to achieve all the gaits and results we demonstrate in this paper.

## 2.4  Energy Consumption-Based Reward

Let's denote the linear velocity as $v$ and the angular velocity as $\omega$, both in the robot's base frame. We additionally define joint torques as $\boldsymbol{\tau}$ and joint angles as $\boldsymbol{q}$ and joint velocities as $\dot{\boldsymbol{q}}$. We define our reward as sum of the following three terms:

$$r = r_{\text{forward}} + \alpha_1 * r_{\text{energy}} + r_{\text{alive}} \tag{3}$$

where,
$$r_{\text{forward}} = -\alpha_2 * |v_x - v_x^{\text{target}}| - |v_y|^2 - |\omega_{\text{yaw}}|^2 \tag{4}$$

$$r_{\text{energy}} = -\boldsymbol{\tau}^T \dot{\boldsymbol{q}}, \tag{5}$$

$r_{\text{forward}}$ rewards the agent for walking straight at the specified speed, $r_{\text{energy}}$ penalizes energy consumption and $r_{\text{alive}}$ is the survival bonus.

We use simple rules to set the hyper-parameters. At a given target linear speed $v_x^{\text{target}}$, the survival bonus $c$ is set to $20 * v_x^{\text{target}}$. We set $\alpha_1 = 0.04$, and $\alpha_2 = 20$ across all settings.

We estimate the unit energy consumption per time step by summing the instantaneous power of the 12 motors by multiplying the torque and the joint velocity at each motor. Also notice that this reward is much more concise than those used in prior works. We discuss the role of minimizing energy consumption in Section 4.4.

## 2.5  Sim to Real Transfer

For simulation to real transfer, we use RMA [17] which consists of the an adaptation module on top of the base policy. During deployment, the adaptation module uses the state history to estimate the vector of extrinsics (which contains environment information) online. Once the base policy is learned using RL, this adaptation module can be trained in simulation itself using supervised learning with the true extrinsics parameters as supervision. The environment perturbations we train on is listed in Table 1.

# 3  Experimental Setup and Training Details

**Hardware and Simulation:**  We use Unitree's A1 robot as our hardware platform [23] and use its A1 URDF [23] to simulate the A1 robot in RaiSim simulator [24]. The full details of terrain and environment setup are provided in supplementary due to space constraints.

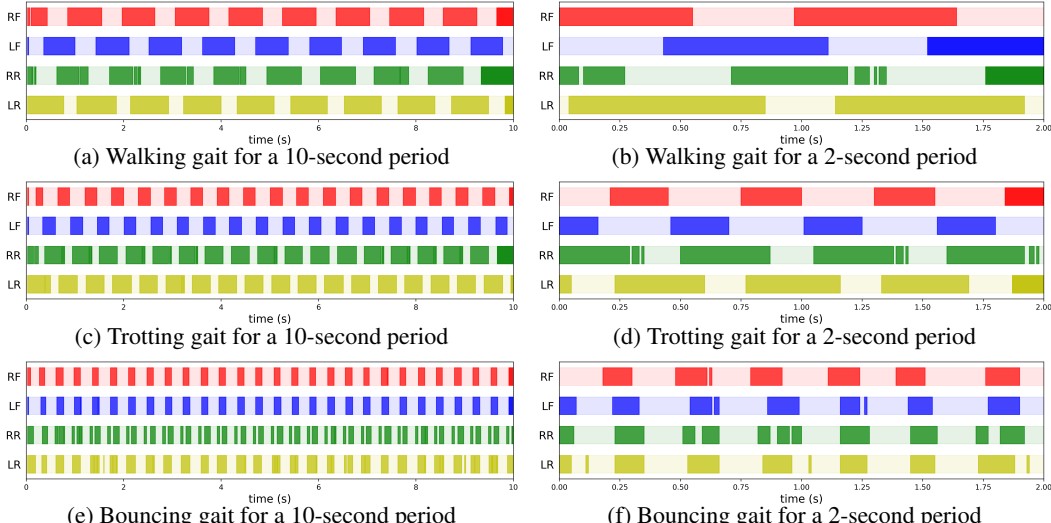

(a) Walking gait for a 10-second period

(b) Walking gait for a 2-second period

(c) Trotting gait for a 10-second period

(d) Trotting gait for a 2-second period

(e) Bouncing gait for a 10-second period

(f) Bouncing gait for a 2-second period

Figure 4: Foot contact plots for walking, trotting and bouncing gait of A1 robot in the real world. Bold color means the corresponding foot is in contact with the ground and the light color means the foot is in the air. (RF: Right-Front foot, LF: Left-Front foot, RR: Right-Rear foot, LR: Left-Rear foot)

**State-Action Space:** The state is 30 dimensional containing the joint positions (12 values), joint velocities (12 values), roll and pitch of the torso and binary foot contact indicators (4 values). The action space is 12 dimensional corresponding to the target joint position for the 12 robot joints. The predicted target joint angles $a = \hat{\mathbf{q}} \in \mathbb{R}^{12}$ is converted to torques $\boldsymbol{\tau}$ using a PD controller with target joint velocities set to 0.

**Environmental Variations:** All environmental variations with their ranges are listed in Table 1. Policies with simple terrain and mild environment variations achieve walking, trotting and bouncing gaits. To deploy our robot in the complex terrain, we use aggressive environment randomization with sharper fractal terrains which gives us the unstructured walking terrain.

**Policy Learning:** The policy is a multi-layer perceptron with 3 layers which takes in the current state $x_t \in \mathbb{R}^{30}$, previous action $a_{t-1} \in \mathbb{R}^{12}$ and outputs 12-dim target joint angles. The hidden layers have 128 units. We train the policy and the environment encoder network using

| Parameters | Normal Perturbation | Aggressive Perturbation |
|---|---|---|
| Friction Coeff. | [0.6, 1.2] | [0.05, 4.5] |
| $K_p$ | [50, 60] | [50, 60] |
| $K_d$ | [0.4, 0.8] | [0.4, 0.8] |
| Payload (kg) | [0.0, 0.5] | [0.0, 6.0] |
| Center of Mass (m) | [-0.15, 0.15] | [-0.15, 0.15] |
| Motor Strength | [0.95, 1.05] | [0.90, 1.10] |
| Re-sample Prob. | 0.02 | 0.02 |

Table 1: Ranges of the environment parameters in simulation. Normal perturbation is used for regular gait emergence. Aggressive perturbation is used for unstructured gait emergence.

PPO [25]. The training runs for 15, 000 iterations with a batch size of 100, 000 split into 4 minibatches and learning rate = 5e−4. We train for a total of 1.5 billion samples which takes roughly 24 hours on a desktop machine with 1 GPU.

## 4    Results and Analysis

### 4.1    Emergence of Locomotion Patterns in Complex Terrain

We analyze the performance of the walking policy on complex terrain with aggressive perturbations in the real world (videos on website). We show two uneven terrain deployments of our method in Figure 3. During the deployment of the robot in the rocky terrain, we see that the robot roughly follows the walk-

| Method | Walk (0.375 m/s) | | Trot (0.9 m/s) | | Bounce (1.5 m/s) | |
|---|---|---|---|---|---|---|
| | Speed (m/s) | Energy | Speed (m/s) | Energy | Speed (m/s) | Energy |
| Ours | 0.353 | 30.7 | 0.970 | 41.2 | 1.439 | 90.7 |
| MPC [6] | 0.360 | 87.0 | 0.953 | 77.2 | 1.482 | 93.9 |

Table 2: We show the actual speeds and energy consumed to move one meter of the our method and convex MPC, which does not directly optimize for energy efficiency, for walking gait at target speed 0.375 m/s, trotting gait at target speed 0.9 m/s, and bouncing gait at target speed 1.5 m/s in simulation. Our method shows the advantages in energy consumption in all gaits.

ing gait while stepping over rocks of different heights which prematurely terminate the swing motion of the leg. Consequently, the gait we observe is an unstructured walking gait. We further analyse this in another deployment on unstable planks which move as the robot tries to cross it (see video).

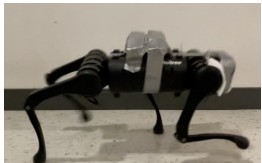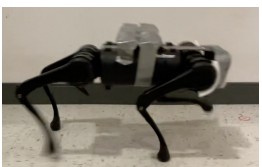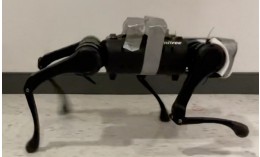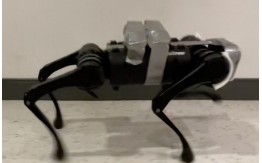

Figure 5: Trotting with 1 kg payload (two bottles of 500 ml water are strapped onto the robot). The 1 kg payload is out of the normal perturbation of environment parameters used in simulation training shown in Table 1. We show qualitative and gait patterns for other two gaits in the appendix. We find that the gait patters are robust and remain the same even in the presence of disturbances.

The foot contact plot for this setup is not periodic and changes depending on the complexity of the terrain. In the video, this can been seen when the front right foot of the robot steps on an unstable plank which moves as a result, but to maintain stability, the robot violates the periodic gait notions automatically.

## 4.2 Emergence of Walking, Trotting and Bouncing

**Simulation Gaits:** We train 3 separate policies for 3 target speeds - 0.375 m/s (low), 0.9 m/s (median) and 1.5 m/s (high) in simulation. We use the same hyper-parameters across all target speeds. We observe that the walking gait emerges at target speed 0.375 m/s, trotting emerges at 0.9 m/s and hopping emerges at 1.5 m/s in simulation. Note that we use RMA [17] (Section 2.5) to transfer our policies to the real world without any fine-tuning. We observe the same foot contact sequence in the real world as in simulation. In Table 2, we compare the performance of our policies with MPC [6] and observe that our walking gait can achieve accurate speed tracking while being 50% more energy efficient than the MPC baseline.

**Real-World Behaviors:** At the low speed (0.375 m/s), the robot demonstrates a *quarter-off* walking gait with the following foot contact sequence: Right-Front (RF), Left-Rear (LR), Left-Front (LF) and Right-Rear (RR). At the median speed (0.9 m/s), two beat trot gait emerges where diagonal legs (RF & LR or LF & RR) are synchronized and hit the ground at the same time. At the high speed (1.5 m/s), the policy converges to bouncing with approximately half the gait cycle as flight phase. Key frames of the 3 gaits are shown in Supplementary and the foot contact plots in Figure 4. We use the foot contact sensor reading from the A1 robot to generate these plots.

We also measured the actual speed of the robot in the real world by taking the average of 3 trials per target speed. In each trial, we log the total distance divided by total time taken to traverse. The average real-world speeds are 0.396 m/s at target speed 0.375 m/s, 0.914 m/s at target speed 0.9 m/s, and 1.714 m/s at target speed 1.5 m/s. The average real-world speeds closely match the target speeds in all settings. The standard deviation of speeds across 3 trials is negligible.

We test the robustness of the emergent gaits by adding a 1 Kg payload on the robot by strapping two bottles with 500 ml water. In Figure 5, we show 4 key frames of trotting gait. Note that the 1 kg payload is outside the range of training perturbations (Table 1), but our policies with the Sim-to-Real Adaptation in Section 2.5 learn to compensate the increase in weight by implicitly predicting changes in dynamics via latent extrinsics. We didn't test bouncing with payload, because the extra 1 kg prevents the human operator from swiftly preempting the robot at high speed by lifting.

**Froude Number Analysis:** In Biomechanics, Froude number $F$ is a scalar metric characterizing gaits of quadrupeds and bipeds [18]. Concretely, $F = v^2/gh$, where $v$ is the linear speed, $g$ is the gravitational acceleration, and $h$ is the height of the hip joint from the ground. Existing research shows that animals with a similar morphology but with different

| Gaits (Froude #) | A1 | Sheep [10] | Horses [12] | Dogs [10] |
|---|---|---|---|---|
| Walk | 0.059 | 0.05 | 0.07 | 0.10 |
| Trot | 0.316 | 0.30 | 0.61 | 0.50 |
| Bounce | 1.110 | 1.10 | 1.70 | 1.73 |

Table 3: We compare the Froude numbers of our emergent gaits with other quadruped animals at their gait transition boundaries. Our policies are closest to sheep in all gaits and to horse in walking gait. Bounce includes canter and gallop.

sizes tend to use the same gait when walking with equal Froude numbers [26, 10]. We calculate the Froude numbers of our robot in the real world for the low, median and high speeds, and compare them with the Froude numbers of quadrupeds in nature – dogs, horses and sheep [12, 10]. We compare the Froude number of animals at the gait transition boundaries and observe that our robot is closest to the sheep in Table 3.

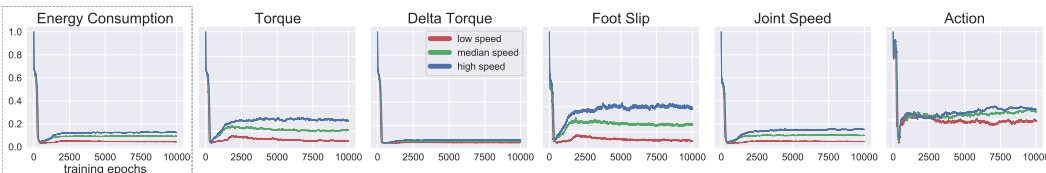

Figure 7: Energy consumption plot during training. Even though our reward function does not include the hand-designed locomotion penalties that are commonly used in prior works (torque, delta torque, foot slip, joint speed, and action), we show that all these penalties are minimized in our training framework as a byproduct of minimizing energy consumption.

## 4.3 Gait Transition via a Velocity-Conditioned Policy

We have shown walking, trotting and bouncing emerged at three different target speeds. To demonstrate foot patterns at a continuous range of velocities, we propose a learning scheme (Figure 6) for a velocity-conditioned policy that enables smooth gait transition when the command velocity changes.

**Naive Multi-Task Training Fails:** We first tried the naive approach to train the velocity-conditioned policy in a multi-task fashion by randomly sampling desired velocities and using the corresponding velocity-conditioned reward (Section 2.4). However, this did not work and the resulting emerged gaits collapse to only two modes: walking and trotting, but trotting at high speeds is not energy-efficient (Figure 2). We believe the reason for failure is difficulty in optimization as the robot is now tasked not only to learn to move forward but also do it by learning different gaits which causes it to collapse.

**Learning via Distillation and RL:** We sidestep this issue via stage-wise distillation approach. We first train our fixed-velocity policies as described in the paper which lead to walking, trotting and bouncing (galloping) gaits. We then treat these policies as experts and collect demonstration data from them to self-supervise and bootstrap the initial training phase of the velocity-conditioned policy. These three policies serve as experts at three velocities at low (0.375 m/s), median (0.9 m/s) and high (1.5 m/s), represented as three-dimensional one-hot vectors, which are fed into the velocity-conditioned policy as input. Expert supervisions along with the RL rewards guide the velocity-conditioned policy

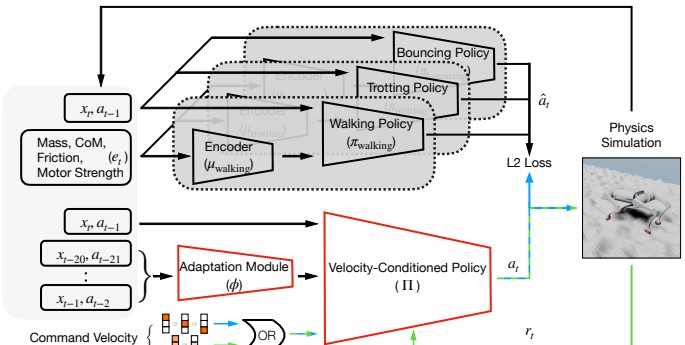

Figure 6: Learning scheme for smooth gait transition. Trainable modules are in red. The blue path indicates forward and backward passes for the L2 loss on action, whereas the green path is for training by RL.

to follow the energy-efficient gaits at the three expert velocities. To learn motor skills and smooth gait transition at intermediate velocities in the continuous range of 0.375 m/s to 1.5 m/s, the learning relies on the velocity-conditioned RL rewards only without expert supervision. To represent a randomly sampled target velocity to feed into the policy, we use an interpolated vector based on the closest two experts' one-hot velocity representations (e.g. 1.2 m/s is represented as [0, 0.5, 0.5] and 0.5 m/s as [0.238, 0.762, 0]). We linearly anneal the L2 supervision loss from the expert supervision (for the three velocity modes) as the training progresses. Towards the end, the learning relies only on the velocity-conditioned RL loss for the entire velocity range. Details are in the supplementary.

**Real-World Smooth Gait Transition:** We test our velocity-conditioned policy in the wild and want to highlight the smooth gait transition happening at different speeds. Video is in the supplementary.

## 4.4 Ablation Studies

The two key components we use during policy training to get reliable walking gaits is an energy based reward function, and a fractal terrain. We posit that these ingredients are minimal, and we demonstrate this by ablating each of them below.

**Minimizing Energy Consumption Also Minimizes Artificial Penalties:** To demonstrate the sufficiency of minimizing energy consumption, we plot the decreasing trend of the artificial penalties during training in Figure 7, even though these artificial penalties are not included in our reward function. These include Torque penalty, delta torque penalty, foot slip penalty, joint speed penalty, and

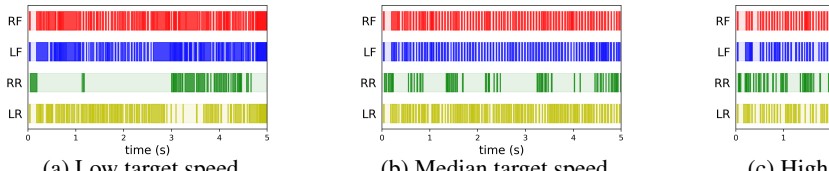

|          | (a) Low target speed | (b) Median target speed | (c) High target speed |
|----------|----------------------|-------------------------|-----------------------|

Figure 8: Resulting gait plots of policies without minimizing energy consumption. All policies exhibit a high-frequency tapping behavior that is not energy-efficient and not transferable to robots in the real world. In (c), the robot in simulation falls off the terrain after 4 seconds.

the action regularization [21, 27, 22, 28, 29]. A high positive correlation between energy consumption and artificial penalties can be observed. All these artificial penalties are minimized in our training framework as a byproduct of minimizing energy consumption. During training, the robot first learns to stand still with a minimal energy consumption (around epoch 500), and then starts moving forward with a slightly higher energy consumption.

**Minimizing Energy Consumption:**    In Figure 8, we show the the resulting gaits of policies trained without the unit energy consumption term in the reward function. All policies show high-frequency tapping behaviors that are not natural and inefficient. The high-frequency joint movements can burn the joint motors in the robot and are not transferable to the real hardware.

**Fractal Terrain during Training:**    We train the policies with energy minimization (Section 2.4) but on a flat terrain without fractal perturbations (Section 2.3). All training trials converge to unnatural and unstable gaits. We found that adding fractal terrain also facilitates a larger foot clearance and robustness which improves real-world hardware deployment. We show the key frames of 2 example unnatural gaits in the supplementary.

## 5   Related Works

**Model-Based Optimization for Gaits:**    Model-based controllers to generate gaits [6] optimize for the stance legs for a given foot contact sequence. They use predefined swing leg motion and show stable gait generation on a real robot system. One class of method is contact-implicit optimization [30, 31, 32], optimizing contact forces along with sequences. However, these gaits do not emerge as an energy-minimizing solution to moving at different speeds. [33] analyzes biped gaits by optimizing energy consumption, but uses a minimal mechanical model assuming point-mass body and massless legs. In contrast, our swing leg motion, stance leg force and contact sequence, all emerge from minimizing energy consumption of moving of a real robot. We take a functional approach to gaits where we achieve them not as predefined movements, but as a consequence of energy minimization.

**Learning for Legged Locomotion and Gaits:**    Data-driven learning for legged locomotion has shown robust controllers for quadrupedal robots [22, 34, 35, 36, 37, 38]. Of these, [34, 22] focus on a controller on complex terrains but do not synthesize and analyze leg patterns of the robot across diverse gaits at different target speeds. [35] demonstrates a learning method to select gait controllers, but the low-level gait controllers are predefined primitives. Concurrent work [21] shows the learned gaits and gait transition, but their method relies on many human priors (e.g. predefined cyclic motion priors), shows trotting at high speed with short periods of flight phase, and requires powerful on-board computation power (Apple M1 chip) to solve online MPC optimization. In graphics, [39] shows low energy also plays a role in generating natural locomotion animations.

## 6   Conclusion

This work demonstrates that energy consumption plays a key role in the emergence of natural locomotion patterns in animals by following an analysis-by-synthesis approach of showing the result in real quadruped robots. The allows generation of the straight line lab gaits, as well as unstructured complex terrain gaits in animals. We perform thorough analysis of gait patterns, show their robustness under disturbances, and propose a learning pipeline for smooth gait transition . Interestingly, the gaits obtained by our robot are most similar to those in horses and sheep – the animals which are close to it in terms of the Froude number metric space.

**Acknowledgments**    This work was supported by the DARPA Machine Common Sense program. DP is partly supported by Good AI research grant. We would like to thank Chris Atkeson for high-quality feedback, and Qingqing Zhao and Shivam Duggal for many video recordings.

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
