# OpenReview forum: "Minimizing Energy Consumption Leads to the Emergence of Gaits in Legged Robots"
_robot-learning.org/CoRL/2021/Conference — CoRL2021 Poster_

### Official Review · Reviewer_CQpm · 2021-07-12

**Originality:** Good
**Technical Quality:** Very Good
**Clarity Of Presentation:** Very Good
**Impact:** 3

**Recommendation:**

Weak Accept: I recommend accepting the paper, but will not argue for my recommendation if the majority of other reviewers have a different opinion.

**Summary:**

This paper presents an analysis that with simple energy minimization and training on fractal terrain can lead to different gaits at different target speeds.

**Issues:**

Questions/Concerns:

1. The claim "Show that minimizing energy consumption is sufficient for the emergence of natural locomotion patterns" is not true, as mentioned in the paper, fractal terrain during training is also necessary.

2. How repeatable are the experiments? e.g., if I train with 10 different random seeds, will all seeds give us the same gait for the same speed?

3. Can we see what the gaits trained on flat ground looks like in the video? I can't get any information from Figure 8 alone.

4. It will be interesting to see the transition between different speeds.

5. Will the authors open source the code? Open source implementation will greatly benefit the community, especially given the affordable price of the robots.

**Reviewer Expertise:**

Very good: Comprehensive knowledge of the area

**Strengths And Weaknesses:**

Pro:

1. Demonstrate that a simple reward (compared to other work that use similar control structure) is able to generate natural gaits that can work well on the real robots.

2. Froude number analysis with number compared with real animals.

Con:

No major flaw. See more comments below.

**Summary Of Recommendation:**

The idea in the paper is straightforward and the paper is well written. The analysis of gait, especially compared to those of animals' using Froude number is interesting.

However, similar result (gait emergence with simple energy minimization) is also demonstrated in Learning Symmetric and Low-Energy Locomotion, by Yu, et al. Although they did not give the throughout analysis of different gait patterns as provided by this paper.

Another impressive part of the paper is the robustness of the controller, although that mainly benefits from the approach in prior work, i.e.,  Learning quadrupedal locomotion over challenging terrain by Lee et al,  and Rma: Rapid motor adaptation for legged robots, by Kumar et al, and is not a contribution of this paper.

Overall, this paper presents a simple approach to train policies for different locomotion gaits for the A1 robot and can be valuable to the community, although the contribution is a bit incremental.

---

> ### Author Response · Authors · 2021-08-24
> **Response to Reviewer # CQpm [part 1/2]**
>
> Dear Reviewer,
>
> Thank you for the insightful feedback! We are pleased to report that we ran the new experiments you suggested and hope that we have addressed all your concerns below. If there are any further questions, please let us know.
>
> > *It will be interesting to see the transition between different speeds.*
> - As per the reviewer’s suggestion, we ran a velocity conditioned experiment and are able to train a single velocity-conditioned policy that can generate different gaits at different desired velocities and achieve transition between different speeds. We elaborate as follows.
>     - **Naive Approach**: We first tried the naive approach reviewer suggested to train the velocity-conditioned policy in a multi-task fashion by randomly sampling desired velocities and using the corresponding velocity-conditioned reward defined in Section 2.4. However, this did not work and the resulting emerged gaits collapse to only two modes: walking and trotting, but trotting at high speeds is not energy-efficient, as shown in Figure 2. We believe the reason for failure is difficulty in optimization as the robot is now tasked not only to learn to move forward but also do it by learning different gaits which causes it to collapse.
>     - **Gait Transition via Distillation**: We sidestep this issue via a stage-wise distillation approach. We first train our fixed-velocity policies as described in the paper which leads to walking, trotting, and bouncing (galloping) gaits. We then treat these policies as experts and collect demonstration data from them to self-supervise and bootstrap the initial training phase of the velocity-conditioned policy. These 3 policies serve as experts at 3 different velocity modes at low (0.375 m/s), median (0.9 m/s) and high (1.5 m/s). We represent the command velocity inputs at these 3 velocity modes at 3 one-hot vectors. To learn motor skills and smooth gait transition at continuous intermediate velocity, we rely on the velocity-conditioned RL rewards and represent the command velocity inputs as interpolations between the 3 velocity modes (e.g. 1.2 m/s is represented as [0, 0.5, 0.5]). We linearly anneal the L2 supervision loss on action space using trajectories from fixed-velocity policies in the beginning, and the velocity-conditioned policy is optimized with only RL loss at the later training phase. Figure summary of this training pipeline is here: https://drive.google.com/file/d/1We2Smh92q0qVWeN66orZyyenKrvoZ4i9/view?usp=sharing.
>     - **Result**: We test our velocity-conditioned policy in the wild. We would like to highlight the smooth gait transition happening at different speeds as seen in the video:  https://drive.google.com/file/d/18GIh1GcafS8Z9b_CYDz7scom9nZnTIu3/view?usp=sharing.
>
> > *However, similar result (gait emergence with simple energy minimization) is also demonstrated in Learning Symmetric and Low-Energy Locomotion, by Yu, et al. Although they did not give the throughout analysis of different gait patterns as provided by this paper.*
> - Thanks for the reference, we will include this paper in the related works. However, there are many differences. They use a more constrained controller where symmetry is imposed by adding a soft loss term, which can speed up training in simulation but no real-world robot performance increase is demonstrated in the paper. Additionally, they use a non-trivial curriculum and a complex reward function, where the hyperparameters are not fixed across different target speeds. We show in 4.3, that many of these hand-designed reward terms can be minimized by only penalizing the energy consumption of the robot on a fractal terrain.
>
> > *How repeatable are the experiments? e.g., if I train with 10 different random seeds, will all seeds give us the same gait for the same speed?*
> - We thank the reviewer for pointing that out. We analyzed the robustness and repeatability of the training procedure by training for multiple trials and the result is available here: https://drive.google.com/file/d/16J_W16BxM3tJjOBxE1unFtpzRuqo1HRA/view?usp=sharing . The plot shows mean performance across 5 seeds and the shaded area shows the standard deviation. As we see from plots, the training converges in all the runs and the standard deviation is very small. In addition, the foot patterns also converge: walk at low speed, trot at median speed, and bounce (gallop) at high speed.
> - We will add these multiple seed experiments in the paper.
>
> (continued in part 2/2 below...)

---

> > ### Author Response · Authors · 2021-08-24
> > **Response to Reviewer # CQpm [part 2/2]**
> >
> > > *Can we see what the gaits trained on flat ground looks like in the video? I can't get any information from Figure 8 alone.*
> > - Here is the video of what happens if we train on a fully flat ground: https://drive.google.com/file/d/1VJd6VK5IpxPJjleq4fPHMBX3kxlJktne/view?usp=sharing.  There are two reasons for failures: (1) The flat terrain does not force the robot to move forward with a certain degree of foot clearance, which is crucial for robots in the wild. (2) Titled and asymmetric gaits are compatible with the unrealistic flat terrain, but any small perturbation on the terrain will lead to falling.
> >
> > > *Another impressive part of the paper is the robustness of the controller, although that mainly benefits from the approach in prior work, i.e., Learning quadrupedal locomotion over challenging terrain by Lee et al, and Rma: Rapid motor adaptation for legged robots, by Kumar et al, and is not a contribution of this paper.*
> > - We are very happy to know that the reviewer found the real robot results impressive. We would like to note that the contribution of this work is orthogonal as well as complementary to RMA. The energy efficient gaits corresponding to different speeds are realistic and we use RMA to show this by using it as a means of sim2real. While it is true that RMA can improve robustness of sim2real transfer, RMA by itself *can not make unnatural gaits transfer* from simulation to real. Hence, generating natural and realistic gaits which are good enough to transfer is still the onus of our paper.
> > - Furthermore, unlike “Learning quadrupedal locomotion over challenging terrain” by Lee et al, we do not use any predefined foot trajectories and real robot data.
> >
> > > *The claim "Show that minimizing energy consumption is sufficient for the emergence of natural locomotion patterns" is not true, as mentioned in the paper, fractal terrain during training is also necessary.*
> > - Yes, the learned behaviour is a consequence of the energy as well as the environment in which the agent is trained. For instance, if the environment is simple, the emergent gaits are structured and if the environment is complex, the resulting gaits are unstructured. We do acknowledge this at several places in the paper but will include it as part of the claim as well to make it crisp. Thanks for the suggestion.
> >
> > > *Will the authors open source the code? Open source implementation will greatly benefit the community, especially given the affordable price of the robots.*
> > - Thank you for the suggestion. We will consider releasing the code. We also note that we have provided all the details necessary to reproduce the paper in the experiments section and are happy to include any additional details which might be missing.

---

> > > ### Comment · Reviewer_CQpm · 2021-08-24
> > > **difference to  Lee et al.**
> > >
> > > This comment is not something that will change my rating, just for argument's sake.
> > >
> > > Lee et al used real robot data to train the actuator model because they showed in their prior work that the actuator on Anymal caused sim-to-real problem. Unless the authors are able to show their method without the actuator network can work on Anymal, their claim about not using real robot data is novel is not convincing.
> > >
> > > predefined foot trajectories:
> > > Their use of predefine foot trajectory is to warm start the policy, their policies in principle can modify the trajectories to assume different gaits. Unless the authors are able to show that using a predefine foot trajectory will make the current framework fail, using a predefine foot trajectory or not doesn't make much difference.

---

> > > > ### Author Response · Authors · 2021-08-25
> > > > **Response to Followup**
> > > >
> > > > Dear Reviewer,
> > > >
> > > > Thank you for your insightful comments regarding the seminal work of Lee et.al. We will discuss this in the related work and also highlight that the objective of our paper is orthogonal as well as complementary to Lee et.al. which can potentially be combined together with our approach (in the same manner as RMA).
> > > >
> > > > We also hope that you like the results of the gait transition experiment we added upon your suggestion. Thank you!

---

> ### Comment · Reviewer_CQpm · 2021-08-31
> **keep my initial rating**
>
> The authors' rebuttal addresses my main concern and I also like the additional speed conditioned policy. I maintain my position on this paper.

---

### Official Review · Reviewer_hrLb · 2021-07-20

**Originality:** Fair
**Technical Quality:** Good
**Clarity Of Presentation:** Good
**Impact:** 3

**Recommendation:**

Weak Accept: I recommend accepting the paper, but will not argue for my recommendation if the majority of other reviewers have a different opinion.

**Summary:**

The paper explores the emergence of different quadruped gaits using reinforcement learning, by maximizing a forward reward and minimizing a lateral movement/rotation and an energy cost.
The agent is trained in simulation with domain randomization, and successfully deployed in the real world using Rapid Motor Adaptation (RMA).

**Issues:**

The authors must clarify the contribution of the paper with respect to RMA.

The claim that minimizing energy consumption is sufficient for the emergence of natural locomotion gaits is also an overstatement given the experimental details from the paper.
It is true that is plays an important regularizer role, but it is not sufficient, as the task reward is encoded in the forward reward + lateral movement/rotation cost.

The result of the Froude number analysis is unclear: it is true that the learned gaits are closest to the sheep, but what can we do with such information?

**Reviewer Expertise:**

Very good: Comprehensive knowledge of the area

**Strengths And Weaknesses:**

Pros:
The paper shows that several quadruped gaits can be successfully learned in simulation and then deployed in the real world  by maximizing a forward reward and minimizing a lateral movement/rotation and an energy cost.
The obtained gaits have better energy efficiency than their MPC counterpart for different target speed.
The analysis shows that other quantities (for instance continuity cost of the torque) are implicitly minimized thanks to the energy term.

Cons:
The main issue with this paper is the unclear contribution with respect to Rapid Motor Adaptation (RMA), which is only briefly mentioned in section 2.5.
In fact, the current paper can be summarized as RMA minus the additional auxiliary rewards (action magnitude, smoothness, foot slip, ...).

The rest of the experimental setting is the same, and the robustness and sim2real transfer can be mainly attributed to RMA methodology:
- domain randomization
- domain adaption (RMA)
- fractal environment


**Summary Of Recommendation:**

Despite good real world results, as the contribution with respect to RMA is unclear, I would recommend this paper for rejection, but I'm willing to improve my rating upon clarification by the authors.

---

> ### Author Response · Authors · 2021-08-24
> **Response to Reviewer # hrLb (with new experiments)**
>
> Dear Reviewer,
>
> Thank you for the comments! We hope that we have addressed all your concerns below. If there are any further concerns, please let us know.
>
> > *The main issue with this paper is the unclear contribution with respect to [...] (RMA), which is only briefly mentioned*
> - The contribution of this work is orthogonal as well as complementary to RMA. The energy-efficient gaits corresponding to different speeds are realistic and we use RMA as a means to show this by using it for sim2real transfer (could have also used domain randomization but RMA works better). While it is true that RMA can improve the robustness of sim2real transfer, RMA by itself *can not make unnatural gaits transfer* from simulation to real. Hence, generating natural and realistic gaits which are good enough to transfer is still the onus of our paper.
> - **Gait Transition via Distillation**: In addition, following suggestions from other reviewers, we ran the velocity conditioned experiment and are able to train a single velocity-conditioned policy that can generate different gaits at different desired velocities and achieve smooth gait transition. We elaborate as follows.
>     - *Naive Approach*: We first tried the naive approach reviewer suggested to train the velocity-conditioned policy in a multi-task fashion by randomly sampling desired velocities and using the corresponding velocity-conditioned reward defined in Section 2.4. However, this did not work and the resulting emerged gaits collapse to only two modes: walking and trotting as the robot tries to learn to both walk forward and learn diverse gaits.
>     - *Gait Transition via Distillation*: We sidestep this issue via a stage-wise distillation approach. We first train our three fixed-velocity policies as described in the paper which leads to walking, trotting, and bouncing (galloping) gaits. We then treat these policies as experts and collect demonstration data from them to self-supervise and bootstrap the initial training phase of the velocity-conditioned policy. We represent the command velocity inputs at these 3 velocity modes at 3 one-hot vectors. To learn motor skills and smooth gait transition at continuous intermediate velocities, we rely on the velocity-conditioned RL rewards and represent the command velocity inputs as interpolations between the 3 velocity modes (e.g. 1.2 m/s is represented as [0, 0.5, 0.5]). We linearly anneal the L2 supervision loss on action space using trajectories from fixed-velocity policies in the beginning, and the velocity-conditioned policy is optimized with only RL loss at the later training phase. Summary figure: https://drive.google.com/file/d/1We2Smh92q0qVWeN66orZyyenKrvoZ4i9/view?usp=sharing.
>     - *Result*: We test our velocity-conditioned policy in the wild and show smooth gait transition happening at different speeds, video:  https://drive.google.com/file/d/18GIh1GcafS8Z9b_CYDz7scom9nZnTIu3/view?usp=sharing.
> - **Comparison to Biological Gaits**: We qualitatively compare our results to biological quadrupeds:
>     - We include a plot of energy efficiency of different gaits at different speeds for our robot which is qualitatively similar to the horse plot from Hoyt et al. 1981.
>     - We include the foot fall patterns or gait plots of all our emergent gaits, which are very close to the gaits observed in biological quadrupeds (see [2] for gaits in horses).
>     - We include a Froude number analysis of our robot and compare it to that of a horse, dog and sheep [10].
>
> > *It is true that [minimizing energy consumption] plays an important regularizer role, but it is not sufficient, as the task reward is encoded in the forward reward + lateral movement/rotation cost.*
> - The standard practice in RL for locomotion community is to heavily shape the reward *in addition* to the task reward. The current wording of the statement was attempting to focus on the difference between them and our method, which only uses energy efficiency in conjunction with task rewards to achieve natural behaviors. The energy has to be minimized in the context of performing some tasks which in this case is moving forward. We will reword this claim to include task rewards to make it precise.
>
> > *The result of the Froude number analysis is unclear: it is true that the learned gaits are closest to the sheep, but what can we do with such information?*
> - We believe our Froude number analysis and qualitative comparison (in Figure 2) will provide insights into biomechanics. It has been an open question in biomechanics and energetics as to how/why do biological quadruped animals display structured gaits [9]. Energy minimization has been shown as a potential candidate but has always been studied in very constrained simple setups by assuming a strict model for the animal (for instance, [16] considers a point-mass model for quadruped animals). In this work, we show that this is indeed true in a model-free framework without assuming any hand-designed model for the quadrupeds.

---

> ### Comment · Reviewer_hrLb · 2021-09-01
> **Final Rating**
>
> I would like to thank the authors for their response. Although they only partially addressed my concerns, I appreciate the re-wording done in the paper and therefore raise my recommendation to "weak accept".

---

### Official Review · Reviewer_gEhV · 2021-07-22

**Originality:** Good
**Technical Quality:** Very Good
**Clarity Of Presentation:** Excellent
**Impact:** 3

**Recommendation:**

Weak Accept: I recommend accepting the paper, but will not argue for my recommendation if the majority of other reviewers have a different opinion.

**Summary:**

This paper explores a hypothesis of gait emergence on a quadruped robot that energy minimization leads to different natural gait patterns such as walking, trot and bouncing gaits depending on the given target walking speed. First, the locomotion policy, which is a mapping from the current state and the previous action represented using a three-layer perceptron, is optimized via model-free reinforcement learning in numerical simulations using PPO. In order to obtain a robust gait policy on a flat surface, the policy is trained on uneven terrains instead of adding simulation noise. A simple reward function is used consisting of the terms associated with the forward velocity and the posture of the robot, penalizing energy consumption, and ground impact. Then, the learned policy in numerical simulations is transferred to a real quadruped robot using RMA. It is demonstrated that a robust walking policy can be obtained without any pre-programming or pre-defined movements for leg motions using the proposed framework on the physical robot. Furthermore, empirically, it is shown that different natural locomotion patterns can emerge depending on the speed of the gait as a consequence of energy minimization. Comparison of the efficiency of gaits obtained in the literature illustrates suggests the advantage of the proposed model-free approach.

**Issues:**

The paper could be more informative if additional technical details that make successful implementation of the learning framework could be provided in the main text of the paper by addressing the following points:

1. How was the reward designed? How did the authors choose each term in the reward function and how were the weights tuned?
2. How was the neural network structure (3-layer perceptron with 128 units) chosen for policy representation? The reviewer wonders whether authors explored any other choices for function approximators.
3. How was the training initialized?
4. In order to demonstrate the learning performance, it is suggested including a learning curve for multiple runs. Was learning always converge and robust?
5. It would be more informative if the reason why the authors did not choose domain randomization in this study.
6. Providing discussions on the advantage of the proposed model-free method over an MPC approach in terms of resultant efficiency would be more informative.
7. Providing detailed discussions on the technical differences in the approaches in comparison to the related work in Section 5 would be more informative.
8. It would be more interesting to see qualitative comparisons of the results to biomechanical studies of animal locomotion from a scientific aspect.


**Reviewer Expertise:**

Very good: Comprehensive knowledge of the area

**Strengths And Weaknesses:**

Strength:
This paper is clearly written and easy to follow. As the supplementary video shows, the obtained policy works remarkably well in the real quadruped robot demonstrating the robustness against perturbations and adaptability to uneven terrain in real environments. In addition, the hardware experiments well-demonstrate the hypothesis raised in this work from an empirical point of view through the proposed reinforcement learning-based optimization approach. The resultant performance is impressive given that the locomotion policy can be obtained via model-free reinforcement learning without any pre-programming or pre-defined leg movements out of a simple energy minimizing reward, or any real world fine-tuning in transferring trained policies from simulations to real environments.

Weakness:
Indeed, the successful application of reinforcement learning to train locomotion policy in generating natural different gaits on a real quadruped robot via energy minimization is highly impressive. First, the idea of minimizing energy for gait discovery has been previously explored in the case of a simplified biped model in the paper [a] listed below. The finding in this study [a] is that different gait patterns such as inverted-pendulum walk and bouncing run depending on the speed of walking can be found via energy minimization by numerically solving an optimal control problem using sequential quadratic programming (SQP). Although there is a difference in the complexity of the problem settings and optimization approaches (either model-based or model-free), the underlying idea seems to be conceptually similar.

[a] Manoj Srinivasan and Andy Ruina, Computer optimization of a minimal biped model discovers walking and running, Vol. 439, 2006, doi:10.1038/nature04113

Second, it is not very clear from the description of the paper how the authors came to choose the learning architecture and the reward function. There is an impression that the proposed framework is somewhat a combination of existing methods. It would be quite surprising to see a relatively simple RL framework with a choice of an energy minimization reward could result in such a complex behavior of the robot in a high-dimensional setting without providing any pre-programming or pre-determined leg movements.


**Summary Of Recommendation:**

The paper is clearly written and well organized. The results with a real robot are very impressive while some of the technical details could be elaborated for the interest of the potential readers and the robot learning community. The paper could be further improved by discussing the motivation and the reason how the authors came to choose the proposed learning architecture, and the actual amount of manual effort in order to make the learning framework successful in such a complex system.

---

> ### Author Response · Authors · 2021-08-24
> **Response to Reviewer # gEhV [part 1/2]**
>
> Dear Reviewer,
>
> Thank you for the insightful feedback! We hope that we have addressed all your concerns below. We also share the new gait transition experiment suggested by the other two reviewers. If there are any further questions, please let us know.
>
> > *The idea of minimizing energy for gait discovery has been previously explored in the case of a simplified biped model in the paper [a] listed below [...] by numerically solving an optimal control problem using sequential quadratic programming (SQP). Although there is a difference in the complexity of the problem settings and optimization approaches (either model-based or model-free)*
> - Thank you for the citation, we will add it to the related work. The introduction of the paper discusses [16], which is a follow-up to citation [a] for quadrupeds. As we discussed in the paper, emergent gaits from energy minimization have been studied for a long time in biomechanics and robotics. We are not the first ones to propose this. Approaches such as [a], [16] primarily focus on conventional gaits in flat terrain whereas the key contribution of our work is that we have the same learning framework which explains both -- conventional structured gaits (walk, trot, gallop) observed in quadrupeds as well as unstructured gaits for uneven and complex terrain. Furthermore, we show this in a fully learning-based setup from scratch without relying on any predefined controllers or demonstrations and then show that these gaits can be deployed feasibly and robustly on a real robot.
>
> > *surprising to see a relatively simple RL framework with a choice of an energy minimization reward could result in such a complex behavior of the robot in a high-dimensional setting without providing any pre-programming or pre-determined leg movements.*
> - To clarify, the learned behavior is a consequence of the energy as well as the environment in which the agent is trained. For instance, if the environment is simple, the emergent gaits are structured and if the environment is complex, the resulting gaits are unstructured.
> - Furthermore, contrary to the common practice of hardcoding foot patterns, we believe that pre-programming behavior would limit the range of behaviors that the agent can find because the search space is more constrained. The lesser the constraints, the richer the behaviors will be and this is also what we empirically demonstrate in this paper, both in simulation and on real robots.
>
> > *How was the reward designed? How did the authors choose each term in the reward function and how were the weights tuned?*
> - Our reward function is motivated by the energy hypothesis in biomechanics. We use the task reward of moving forward, which encourages moving forward straight and penalizes lateral movements and angular yaw speed, a standard in RL for locomotion tasks.
> - For tuning the reward coefficient, we found that increasing the energy term to be high leads to the robot standing in place, or learning to fall forward, whereas a very low energy reward leads to unnatural gaits. The tuning of the reward terms took roughly 2 weeks on a regular desktop computer with 1 GPU. The rewards were tuned to make sure that the energy reward is high enough without overpowering the forward reward. Note that it could be done significantly faster if one has access to a large compute cluster.
>
> > *How was the neural network structure [...] chosen for policy representation? The reviewer wonders whether authors explored any other choices for function approximators.*
> - Thank you, we will add more detail about the architectural and implementation choices in the paper. We discuss some of them here.
>     - *Architecture*: The deep network architecture we use is similar to what is commonly used in reinforcement learning papers for control, e.g., [Hwangbo et al. Science Robotics’19], [Lee et al. Science Robotics’20], [Kumar et al. RSS’21] etc. We also experimented by making slight variations to the policy by using more layers and more neurons, but the performance didn’t seem to change much.
>     - *Observation and action space*: The observation space was chosen to match the reliable sensors available onboard the robot, and the action space was chosen to be the target joint position which itself is tracked by a PD controller. This yielded more stable policies.
>     - *Function Approximators*: We considered only neural networks as function approximators but it would be interesting future work to see how other function approximators perform in this setup.
>
> > *How was the training initialized?*
> - Our neural networks were initialized randomly by using the orthogonal initialization from [Saxe et al., ICLR 2014]. Since we do not use any curriculum, all environment variations are uniformly sampled from fixed ranges during the training phase. The robot in the simulation is initialized to standing at a randomly selected position on the fractal terrain.
>
> (continued in part 2/2 below...)

---

> > ### Author Response · Authors · 2021-08-24
> > **Response to Reviewer # gEhV [part 2/2]**
> >
> > > *In order to demonstrate the learning performance, it is suggested including a learning curve for multiple runs. Was learning always converge and robust?*
> > - Thank you for pointing it out. We analyzed the robustness and repeatability of the training procedure by training for multiple trials and the result is available here: https://drive.google.com/file/d/16J_W16BxM3tJjOBxE1unFtpzRuqo1HRA/view?usp=sharing . The plot shows mean performance across 5 seeds and the shaded area shows the standard deviation. As we see from plots, the training converges in all the runs and the standard deviation is very small. In addition, the foot patterns also converge: walk at low speed, trot at median speed, and bounce (gallop) at high speed.
> >
> > > *It would be more informative if the reason why the authors did not choose domain randomization in this study.*
> > - We considered two choices for the sim2real transfer piece of our work. One was domain randomization [Tobin et.al. 2017] and the other was RMA [Kumar et.al. 2021]. We did not choose domain randomization because the RMA paper shows significantly superior performance compared to domain randomization (see “Robust” policy in Table 2).
> > - Finally, we note that the choice of domain randomization or RMA is orthogonal and complementary to the key contribution of the paper which is to show that energy efficiency with task rewards yields the conventional notion of gaits at different speeds.
> >
> > > *Providing discussions on the advantage of the proposed model-free method over an MPC approach in terms of resultant efficiency would be more informative.*
> > - The Convex MPC approach uses a predefined swing motion that follows a bezier curve or a sine wave with a fixed clearance. However, such a motion is not necessarily energy-efficient. In our framework, we don’t predefine a leg movement and let the policy learn everything from experiential data in simulation. Consequently, our proposed method leads to superior energy efficiency which is not surprising given that our rewards contain energy terms.
> >
> > > *Providing detailed discussions on the technical differences in the approaches in comparison to the related work*
> > - Thank you for suggestion. We will edit related work and summarize key points here.
> > - Most model-based optimization methods for gaits use predefined leg motions and leg clearance, whereas our method only encourages minimal energy consumption without explicit constraints on leg motions. Among learning-based literature, J. Lee et al. [33] focus on a walking controller on complex terrains but do not analyze the leg patterns of the robot across different target speeds. X. Da et al. [34] demonstrate a learning method to select gait controllers, but the low-level gait controllers are predefined primitives. Concurrent work Y. Yang et al. [21] show the learned gaits and gait transition, but their method relies on many human priors (e.g. predefined cyclic motion priors), shows only 2 gaits where the robot keeps trotting at high speed with short periods of flight phase, and requires powerful onboard computation power to solve online MPC optimization (Apple M1 chip).
> >
> > > *It would be more interesting to see qualitative comparisons of the results to biomechanical studies of animal locomotion from a scientific aspect.*
> > - We fully agree. We have the following three analyses to compare our robot to animals and are open to including any additional suggestions the reviewer has in mind to make this analysis stronger.
> >     1. We include a plot of energy efficiency of different gaits at different speeds for our robot which is qualitatively similar to the horse plot from Hoyt et al. 1981 (“Gait and the energetics of locomotion in horses”).
> >     2. We include the footfall patterns or gait plots of all our emergent gaits, which are very close to the gaits observed in biological quadrupeds (see [2] for gaits in horses).
> >     3. We include a Froude number analysis of our robot and compare it to that of a horse, dog and sheep; and observe that our robot resembles the horse and sheep most closely [10].
> >
> > ### New Experiment: Gait Transition
> > - Upon suggestion from other reviewers, we have run a follow-up to obtain a single velocity-conditioned policy that can demonstrate all the gaits as the speed changes. We now have a single velocity-conditioned policy (in contrast to three different speed policies) that automatically switches to the correct gait at different speeds and also demonstrates smooth gait transitions during the switch.
> > - Video result: https://drive.google.com/file/d/18GIh1GcafS8Z9b_CYDz7scom9nZnTIu3/view?usp=sharing
> > - More details in the common reply at the top as well as in this summary: https://drive.google.com/file/d/1We2Smh92q0qVWeN66orZyyenKrvoZ4i9/view?usp=sharing

---

> ### Comment · Reviewer_gEhV · 2021-09-01
> **Final rating**
>
> I would like to thank the authors for their response. The authors have addressed the issues in a satisfactory manner. I would like to keep the initial recommendation.

---

### Official Review · Reviewer_h727 · 2021-07-24

**Originality:** Fair
**Technical Quality:** Good
**Clarity Of Presentation:** Good
**Impact:** 3

**Recommendation:**

Weak Accept: I recommend accepting the paper, but will not argue for my recommendation if the majority of other reviewers have a different opinion.

**Summary:**

This paper presents a energy-based end-to-end reinforcement learning approach to realize natural locomotion gaits at different speeds in real quadruped robots. The authors show empirically that energy minimization criterion is enough to realize structured locomotion gaits at different speeds. The gaits show different behaviors for different speeds and match the behavior and energy profile of animals in nature. Finally, the authors validate their approach in both simulation and hardware showing effective speed tracking and robustness against challenging terrains.


**Issues:**

- Bullet 1 and 4 of the main contributions are basically the same.
- Bullet 3 in main contributions is not really a contribution of this paper, but the result of ARM [17].
- Why extra penalties are added when doing uneven terrain with unstructured gaits?

Minor comments:
- According to section 1 in the appendix, the simulation time step is 0.025 seconds while the controller frequency is 100 Hz, which means the controller time step is 0,01 seconds. This results in the control actions being faster than the simulation itself, which adds unnecessary computational cost to the controller.
- There is an error in the dimension of the state space in the appendix. It says 32 when it is 30.
- Move Table 2 to page 6.


**Reviewer Expertise:**

Very good: Comprehensive knowledge of the area

**Strengths And Weaknesses:**

Strengths
-Sometimes less is better. A simple approach such as only using energy consumption is shown to be enough to render natural locomotion gaits without using hand-coded sequences or reference trajectories.
-The method presents an end-to-end model-free reinforcement learning approach that is simple but effective and does not require additional dynamic model information of the robots, which makes the approach general for different quadruped robots.

Weaknesses
-It is hard to differentiate where the robustness of the method comes from since ARM is used on top of the proposed method to realize effective sim-to-real transfer.
-The claim that energy minimization leads to the emergence of structured locomotion gait patterns in flat terrains and unstructured gaits in complex terrains may be imprecise and hard to demonstrate.  In particular, because additional rewards are added for uneven terrain with unstructured gaits and more aggressive environment randomization limits.
-Different training sessions are needed for tracking different speed and therefore to emerge different behaviors.



**Summary Of Recommendation:**

Although the paper presents  good results in terms of realizing energy efficient gaits in both simulation and hardware, it is hard to identify the actual reason for the success of the method, specially in terms of the robustness and sim-to-real transfer. Also, there are some important observations in the contributions, as some of them may not really correspond to the implementation of the proposed method but an addition of a previous method on top of the proposed one.
I would recommend to include the desired velocity as part of the state so that only one trained policy can realize different behaviors for walking at different speeds.

---

> ### Author Response · Authors · 2021-08-24
> **Response to Reviewer # h727 [part 1/2]**
>
> Dear Reviewer,
>
> Thank you for the insightful feedback! We are pleased to report that we finished the new experiments you suggested and hope that we have addressed all your concerns below. If there are any further questions, please let us know.
>
> > *I would recommend to include the desired velocity as part of the state so that only one trained policy can realize different behaviors for walking at different speeds.*
> - As per the reviewer’s suggestion, we ran the velocity conditioned experiment and are able to train a single velocity-conditioned policy that can generate different gaits at different desired velocities and achieve smooth gait transition. We elaborate as follows.
>     - **Naive Approach**: We first tried the naive approach reviewer suggested to train the velocity-conditioned policy in a multi-task fashion by randomly sampling desired velocities and using the corresponding velocity-conditioned reward defined in Section 2.4. However, this did not work and the resulting emerged gaits collapse to only two modes: walking and trotting, but trotting at high speeds is not energy-efficient, as shown in Figure 2. We believe the reason for failure is difficulty in optimization as the robot is now tasked not only to learn to move forward but also do it by learning different gaits which causes it to collapse.
>     - **Gait Transition via Distillation**: We sidestep this issue via a stage-wise distillation approach. We first train our fixed-velocity policies as described in the paper which leads to walking, trotting, and bouncing (galloping) gaits. We then treat these policies as experts and collect demonstration data from them to self-supervise and bootstrap the initial training phase of the velocity-conditioned policy. These 3 policies serve as experts at 3 different velocity modes at low (0.375 m/s), median (0.9 m/s) and high (1.5 m/s). We represent the command velocity inputs at these 3 velocity modes at 3 one-hot vectors. To learn motor skills and smooth gait transition at continuous intermediate velocity, we rely on the velocity-conditioned RL rewards and represent the command velocity inputs as interpolations between the 3 velocity modes (e.g. 1.2 m/s is represented as [0, 0.5, 0.5]). We linearly anneal the L2 supervision loss on action space using trajectories from fixed-velocity policies in the beginning, and the velocity-conditioned policy is optimized with only RL loss at the later training phase. Figure summary of this training pipeline is here: https://drive.google.com/file/d/1We2Smh92q0qVWeN66orZyyenKrvoZ4i9/view?usp=sharing
>     - **Result**: We test our velocity-conditioned policy in the wild. We would like to highlight the smooth gait transition happening at different speeds as seen in the video:  https://drive.google.com/file/d/18GIh1GcafS8Z9b_CYDz7scom9nZnTIu3/view?usp=sharing
>
> > *Why extra penalties are added when doing uneven terrain with unstructured gaits?*
> - To clarify, in the uneven terrains, we add extra penalties like minimizing ground impacts to explicitly reduce the risks of hardware wearing because complex terrain can otherwise damage the hardware quickly. However, these extra penalties are *not* responsible for the unstructured gait emergence in complex uneven terrain.
> - To show this, we re-ran the uneven complex terrain scenario without using any penalty term and just keeping the energy terms, same as in the simpler terrain setting. We found that this does not affect the result in terms of gaits still leading to the emergence of unstructured foot patterns on complex terrains.  Please see video results here and we hope this clarifies the reviewer's concern: https://drive.google.com/file/d/1YQigwdebrf5IoyINJ-E-b14NqKIxQRCS/view?usp=sharing
>
> > *According to section 1 in the appendix, the simulation time step is 0.025 seconds while the controller frequency is 100 Hz, which means the controller time step is 0,01 seconds. This results in the control actions being faster than the simulation itself, which adds unnecessary computational cost to the controller.*
> - This is a typo on our end. We apologize for this. The simulation time step should be 0.0025s, and each target joint position is tracked for 4 times using a PD controller, giving us a control frequency of 0.01s. We will fix this in the paper.
>
> > *Bullet 1 and 4 of the main contributions are basically the same.*
> - We wish to clarify that Bullet 1 refers to the main result of the paper while Bullet 4 is referring to the in-depth ablation analysis of what different choices matter in the energy minimization-based locomotion. If the reviewer feels strongly about merging them, we are open to doing so.
>
>
> (continued in part 2/2 below...)

---

> > ### Author Response · Authors · 2021-08-24
> > **Response to Reviewer # h727 [part 2/2]**
> >
> > > *Bullet 3 in main contributions is not really a contribution of this paper, but the result of RMA [17].*
> > > *It is hard to differentiate where the robustness of the method comes from since RMA is used on top of the proposed method to realize effective sim-to-real transfer.*
> > - The contribution of this work is orthogonal as well as complementary to RMA. The energy-efficient gaits corresponding to different speeds are realistic and we use RMA to show this by using it as a means of sim2real. While it is true that RMA can improve the robustness of sim2real transfer, RMA by itself *can not make unnatural gaits transfer* from simulation to real. Hence, generating natural and realistic gaits which are good enough to transfer is still the onus of our paper.
> >
> > > *There is an error in the dimension of the state space in the appendix. It says 32 when it is 30.*
> > - Thank you. We will fix this.
> >
> > > *Move Table 2 to page 6.*
> > - Thank you for the suggestion.

---

### Author Response · Authors · 2021-08-24
**[To All] Brief Summary of Rebuttal**

We thank the reviewers for their valuable feedback. We are glad that all the reviewers liked the real robot experiments in the paper, and found the approach “general, “simple but effective” (reviewer h727) and the results “remarkable”, “impressive” (reviewer H7pc, gEhV). We’re pleased to report that we’ve finished the additional baselines or experiments suggested by the reviewers. We made detailed responses directly to each review. Here, we recap only the main points from reviewers and the resulting experimental findings.

### Key contribution
1. We consider the energy hypothesis for the emergence of natural locomotion and take the analysis by synthesis approach to show that minimizing energy consumption is sufficient for the emergence of natural locomotion patterns in both flat as well as complex terrains at different speeds without relying on demonstrations or predefined motion heuristics.
2. We further qualitatively compare our results to biological quadrupeds --
    - We include a plot of energy efficiency of different gaits at different speeds for our robot which is qualitatively similar to the horse plot from Hoyt et al. 1981.
    - We include the foot fall patterns or gait plots of all our emergent gaits, which are very close to the gaits observed in biological quadrupeds (see [2] for gaits in horses).
    - We include a Froude number analysis of our robot and compare it to that of a horse, dog, and sheep [10].
3. We deploy our gaits on the real hardware to show that the behaviors we see in the simulation are realistic and natural enough to be realizable on the real hardware.
4. *Gait Transition*: Following suggestions from multiple reviewers, we implement a stage-wise distillation approach to obtain a single velocity-conditioned policy (in contrast to three different speed policies) that automatically switches to the correct gait at different speeds and also demonstrates smooth gait transitions during the switch. See video [here](https://drive.google.com/file/d/18GIh1GcafS8Z9b_CYDz7scom9nZnTIu3/view?usp=sharing) and more details below.

### New Experiments

1. **Velocity-Conditioned Policy & Gait Transition [Reviewer # h727, Reviewer # CQpm]**: Following suggestions from reviewers, we ran the velocity conditioned experiment and are able to train a single velocity-conditioned policy that can generate different gaits at *continuously* varying velocities and achieve *smooth* gait transition. We elaborate below.
    - **Naive Approach**: We first tried the naive approach reviewer suggested to train the velocity-conditioned policy in a multi-task fashion by randomly sampling desired velocities and using the corresponding velocity-conditioned reward defined in Section 2.4. However, this did not work and the resulting emerged gaits collapse to only two modes: walking and trotting as the robot tries to learn to both walk forward and learn diverse gaits.
    - **Gait Transition via Distillation**: We sidestep this issue via a stage-wise distillation approach. We first train our three fixed-velocity policies as described in the paper which leads to walking, trotting, and bouncing (galloping) gaits. We then treat these policies as experts and collect demonstration data from them to self-supervise and bootstrap the initial training phase of the velocity-conditioned policy. We represent the command velocity inputs at these 3 velocity modes at 3 one-hot vectors. To learn motor skills and smooth gait transition at continuous intermediate velocities, we rely on the velocity-conditioned RL rewards and represent the command velocity inputs as interpolations between the 3 velocity modes (e.g. 1.2 m/s is represented as [0, 0.5, 0.5]). We linearly anneal the L2 supervision loss on action space using trajectories from fixed-velocity policies in the beginning, and the velocity-conditioned policy is optimized with only RL loss at the later training phase. Summary figure: https://drive.google.com/file/d/1We2Smh92q0qVWeN66orZyyenKrvoZ4i9/view?usp=sharing.
    - **Result**: We test our velocity-conditioned policy in the wild and show smooth gait transition happening at different speeds, video:  https://drive.google.com/file/d/18GIh1GcafS8Z9b_CYDz7scom9nZnTIu3/view?usp=sharing.

2. **Robustness and Repeatability of Learning Procedure [Reviewer # gEhV, Reviewer # CQpm]**: Following suggestions from reviewers, we analyze the robustness and repeatability of the training procedure by training for multiple trials and the result is available here: https://drive.google.com/file/d/16J_W16BxM3tJjOBxE1unFtpzRuqo1HRA/view?usp=sharing . The plot shows mean performance across 5 seeds and the shaded area shows the standard deviation. As we see from plots, the training converges in all the runs and the standard deviation is very small. In addition, the foot patterns also converge: walk at low speed, trot at median speed, and bounce (gallop) at high speed.

Please refer to more details in the direct responses to individual reviewers.

---

### Meta-Review · Area_Chair_H7pc · 2021-08-11

**Recommendation:** Accept (Poster)
**Confidence:** 4

**Metareview:**

This paper proposes to train locomotion controllers using reinforcement learning with a simple reward function for energy efficiency. Different locomotion gaits emerge automatically at different locomotion speeds, which is consistent with the findings in animal motor control. All reviewers appreciate the impressive robot experiments. The rebuttal cleared the original concern about its delta from the RMA paper. The revision of the paper and new experiments have sufficiently addressed reviewers' questions about details of the learning setup and the repeatability of experiments. In addition, the new results on gait transitions further improved the quality of the paper.

---

> ### Author Response · Authors · 2021-08-24
> **Response to Meta Reviewer # H7pc (with new experiments)**
>
> Dear Area Chair,
>
> We thank you and all the reviewers for appreciating our thorough real-world results. We have clarified all the reviewers' questions in direct responses and post a summary response separately as well. We are pleased to report that we finished all the experiments that the reviewers suggested and hope to have addressed all their concerns.
>
> We note that the objective of this work is orthogonal to RMA and its contribution is complementary and below we further address your concern about the delta between this paper and the RMA paper in detail.
>
> > *the delta between this paper and the RMA paper*?
>
> - **Conceptual/Ideological Difference**: It has been an open question in biomechanics and energetics as to how/why do biological quadruped animals display structured gaits [9]. Energy minimization has been shown as a potential candidate but has always been studied in very constrained simple setups by assuming a strict model for the animal (for instance, [16] considers a point mass model for quadruped animals). In this work, we show that this is indeed true in a model-free framework without assuming any hand designed model for the quadrupeds.
>     - RMA is only used as a choice to perform sim2real transfer (could have also used domain randomization but RMA works better). While it is true that RMA can improve robustness of sim2real transfer, RMA by itself *can not make unnatural gaits transfer* from simulation to real. Hence, generating natural and realistic gaits which are good enough to transfer is still the onus of our paper.
> - **Comparison to Biological Gaits**: We further qualitatively compare our results to biological quadrupeds --
>     - We include a plot of energy efficiency of different gaits at different speeds for our robot which is qualitatively similar to the horse plot from Hoyt et al. 1981.
>     - We include the foot fall patterns or gait plots of all our emergent gaits, which are very close to the gaits observed in biological quadrupeds (see [2] for gaits in horses).
>     - We include a Froude number analysis of our robot and compare it to that of a horse, dog and sheep [10].
> - **New Experiment: Gait Transition via Distillation**: Upon suggestion from multiple reviewers, we have run a follow up to obtain a single velocity-conditioned policy that can demonstrate all the gaits as the speed changes. We now have a single velocity-conditioned policy (in contrast to three different speed policies) that automatically switches to the correct gait at different speeds and also demonstrates smooth gait transitions during the switch. We are pleased to share that result here: https://drive.google.com/file/d/18GIh1GcafS8Z9b_CYDz7scom9nZnTIu3/view?usp=sharing . More details about this are available in the common reply to all as well as in this summary method figure here: https://drive.google.com/file/d/1We2Smh92q0qVWeN66orZyyenKrvoZ4i9/view?usp=sharing

---

### Decision · Program_Chairs · 2021-09-13

**Decision:**

Accept (Poster)

**Comment:**

This paper proposes to train locomotion controllers using reinforcement learning with a simple reward function for energy efficiency. Different locomotion gaits emerge automatically at different locomotion speeds, which is consistent with the findings in animal motor control. All reviewers appreciate the impressive robot experiments. The rebuttal cleared the original concern about its delta from the RMA paper. The revision of the paper and new experiments have sufficiently addressed reviewers' questions about details of the learning setup and the repeatability of experiments. In addition, the new results on gait transitions further improved the quality of the paper.